# Effects of a Post-Weld Heat Treatment on the Mechanical Properties and Microstructure of a Friction-Stir-Welded Beryllium-Copper Alloy

**Yeongseok Lim** [1,2] [ID]**, Kwangjin Lee** [2,]***and Sangdon Moon** [1]

[1]   Department of Mechanical Design Engineering, Chonbuk National University, Jeonju 54896, Korea;
      dudtjr1215@gmail.com (Y.L.); msd@jbnu.ac.kr (S.M.)
[2]   Carbon & Light Materials Application R&D Group, Korea Institute of Industrial Technology, 222, Palbok-ro,
      Deokjin-gu, Jeonju-City 54853, Korea
***   Correspondence: kjlee@kitech.re.kr; Tel.: +82-63-210-3711

**Abstract:** This paper investigated the microstructure and mechanical properties of a friction-stir-welded beryllium-copper alloy, which is difficult to weld with conventional fusion welding processes. Friction stir welding (FSW) was successfully conducted with a tungsten-carbide (WC) tool. Sound joints without defects were obtained with a tool rotational speed of 700 RPM and tool travel speed of 60 mm/min. A post-weld heat treatment (PWHT) of the FSW joints was performed to analyze the evolution of the microstructure at 315 °C for a half, one, two, three, four, five and eight hours, respectively. The microstructures of the joints were observed using an optical microscope (OM), a scanning electron microscope (SEM) and a transmission electron microscope (TEM). Observed softening of microstructure is suggested to be due to the dissolution of the strengthening precipitates during the FSW process, whereas the strength of the joints was recovered via the formation of the CuBe (γ′) phase during the post-weld heat treatment. However, the strength was decreased upon an excessive post-weld heat treatment exceeding three hours. It is considered that the formation of the γ phase and the coarse γ′ phase contributed to the reduction in the strength.

**Keywords:** friction stir welding; beryllium-copper alloy; mechanical properties; microstructure; post-weld heat treatment

---

## 1. Introduction

Cu alloys have been widely used in the aerospace, transportation and electric power industries due to their reasonable strength, excellent conductivity and good corrosion resistance [1]. Moreover, giga-grade high strength beryllium-copper alloy is used to manufacture several components, such as anti-galling cylinders for undersea cable communication system repeater housings, undersea pressure vessels, valves and gimbals, as well as connectors and drill components due to the high strength and hardness of this material and its excellent fatigue, corrosion and wear resistance capabilities [1,2]. Beryllium-copper alloys can be welded using conventional fusion welding methods; however, problems such as the formation of inclusions, blow holes, porosity and solidification cracking in the heat-affected zone (HAZ) can arise. However, this alloy has poor weldability and can induce softening when welded owing to the dissolution of strengthening precipitates [1,3]. Another major concern is the lack of suitable flux/core wire materials for beryllium-copper alloys [3].

A previous study has shown that the copper alloy joints fabricated by laser beam welding (LBM), which is most generally used welding method, show several challenges. For copper alloys with high thermal conductivity and high laser beam absorption, process stability and high laser power are

required as a solution of spattering and high porosity problem [4,5]. Furthermore, when welding thicker plates of 3 mm or more, the higher laser power and the lower welding speed are required; however, defects, spatter and fluctuations due to unstable molten weld pool can be easily formed [6,7]. Despite many laser welding techniques suggested to improve the weldability of copper alloys, the complicated devices used, high cost, laborious set-up and the possible impurities inclusions limits their widespread industrial applications [8]. Especially in case of the welding of beryllium copper alloy, the laser beam welding butt joint of the 0.2 mm of thickness of beryllium copper plate shows sound joint strength as 90% of base metal; however, the liquation crack in HAZ involved in fracture was observed [9]. In solid state joining method, Lap joint of beryllium-copper alloy was fabricated using diffusion brazing with filler metal of Ag without defect at 750 °C for 1200 s. However, the tensile strength of the joint was comparatively low value as 173 MPa [10]. Thus, in order to fabricate sound thick beryllium copper alloy joint without defect caused by melting state, new welding methods are needed to overcome the above limits.

Friction stir welding is a solid-state joining process invented by The Welding Institute (TWI) in the UK [11]. It is an effective method due to the use of a low heat input welding process and eliminates melting and solidification associated problems [12], such as liquidation and solidification cracking [13]. It was well known that the FSW process produces high-quality welded region with a homogeneously refined microstructure and better mechanical properties than those yielded by conventional welding processes [14]. In particular, FSW is an effective technique for joining stain-less steel due to advantages such as low distortion and residual stress [15]. In recent days, studies about high strength steel [16], advanced high strength steel [17] and Ti-alloys [18] were performed. Murugan et al. showed that friction stir welding condition can get sound joints without defects in copper and bronze plates [19]. Sun et al. reported for the formation of denser twins in the stir zone during friction stir welding [20]. Guoliang et al. analyzed the precipitation behavior of Cu-2.0Be alloys in detail, but did not consider the FSW process [21].

In the present study, the authors tried to conduct friction stir welding of Giga-grade high strength beryllium-copper alloy plates and sound joints were successfully manufactured. Post-weld heat treatment (PWHT) was performed to improve the mechanical properties of the joints and the microstructural evolution was investigated during the PWHT. Based on the results, the authors discussed about the relationship between microstructure and mechanical properties during FSW and consequent PWHT. To conclude, the authors revealed the mechanism for dissolution and re-precipitation of strengthening $\gamma'$ precipitates (CuBe) during FSW and consequent PWHT. In addition, this paper optimized the PWHT condition can obtain acceptable mechanical properties.

## 2. Materials and Methods

### 2.1. Materials and FSW Conditions

The material selected in this work is a 3-mm-thick commercial beryllium-copper alloy forged plate, referred to here as "C17200". The chemical composition of the plate is given in Table 1. These plates, with a length of 260 mm and a width of 120 mm, were bead-on-plate welded. Based on the preliminary welding test at various conditions as 600, 700, 800 RPM of a tool rotational speed and 50, 60, 70 mm/min of a tool traveling speed to figure out sound joint condition without impact on the friction stir welding tool and micro defects such as tunnel, crack and void of the joint, Friction stir welding was conducted at a tool rotational speed of 700 RPM and a tool traveling speed of 60 mm/min with an angular advance of 2 degrees as a standard condition. In this study, as shown in Figure 1, a 2.7-mm-long tungsten carbide (WC) tool with a hemispherical probe (6 mm diameter) and a concave shoulder (15 mm diameter) was selected.

## 2.2. Methods and Corresponding Conditions of Analysis

Figure 2 shows schematic representations of tensile and Charpy test specimens. This figure also shows the geometry of the tensile and Charpy testing following ASTM: E8 (gauge length: 40 mm) and KS B 0809 standard, respectively. The specimens were cut perpendicular to the welding direction, with the stir zone (SZ) being centered within the gage length and notch. The FSW specimens were subjected to PWHT at 315 °C for durations ranging from a half hour to eight hours. A salt bath furnace was used for the PWHT process. The heat-treated specimens were subsequently cooled to room temperature. Figure 3 shows the solvus temperature and age-hardening temperature of the precipitates in a phase diagram of the beryllium-copper alloy. The post-weld heat-treatment temperature was determined to be 315 °C after referring to the phase diagram and to previous study results [1,2,15–17]. Figure 4 shows a process flow chart describing the FSW and PWHT condition. The microstructure of the transverse section of the FSW specimens was observed by OM using an Olympus (GX51, Tokyo, Japan) and a field emission scanning electron microscope (FE-SEM, JEOL, JSM-700F, Tokyo, Japan). The observed samples were prepared according to the standard procedure for specimen preparation, including grinding, polishing and etching. Surfaces were observed after immersion for 20 s using 40 vol% of $HNO_3$ and 60 vol% of a $CH_3OH$ solution. Precipitation behavior in the SZ following PWHT was investigated by a high-resolution transmission electron microscope (HR-TEM, JEOL, JEM-2010, Tokyo, Japan) at 200 kV and a cs-corrected-field emission transmission electron microscope (CS-corrected-FE-TEM, JEOL, JEM-ARM-200F, Tokyo, Japan) at 200 kV. The thin foils used for the TEM observation were obtained from the SZ center and from base metal in the plane perpendicular to the welding direction. They were prepared by twin-jet electro polishing in 40 vol% of $HNO_3$ and 60 vol% of a $CH_3OH$ solution at −20 °C. For the microhardness measurements, the samples were cut from the SZ center. Microhardness levels were measured on the welding direction plane of the SZ using a Vickers microhardness test machine (Daekyung Tech, DTR-300N, Incheon, Korea) with a load of 300 g and a dwell time of 10 s. The hardness value was determined by taking an average of five readings and excluding the maximum and minimum values. The tensile test (Instron, 8801, Norwood, MA, USA) was conducted at room temperature at a strain rate of 1.0 mm/min, and the tensile strength values under various PWHT conditions were determined by taking the mean of three test results. The yield strength was determined at an offset of 0.2 percentage.

**Table 1.** Chemical composition of the beryllium-copper alloy used in this work.

| Element | Be | Si | Al | Cu |
|---------|-----|-----|-----|------|
| **Mass%** | 1.9 | 0.2 | 0.2 | Bal. |

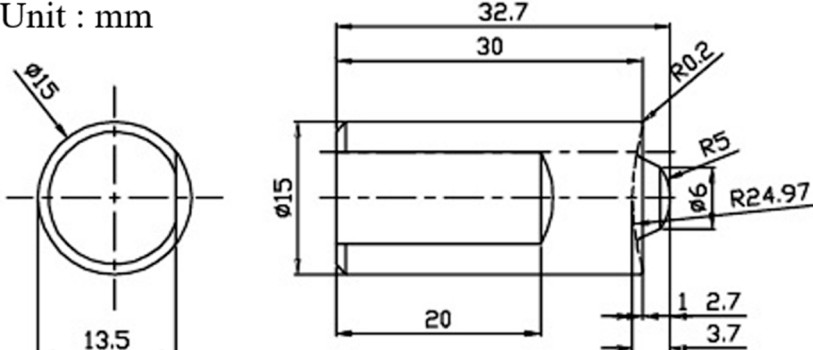

**Figure 1.** Illustraion of Tool geomatric.

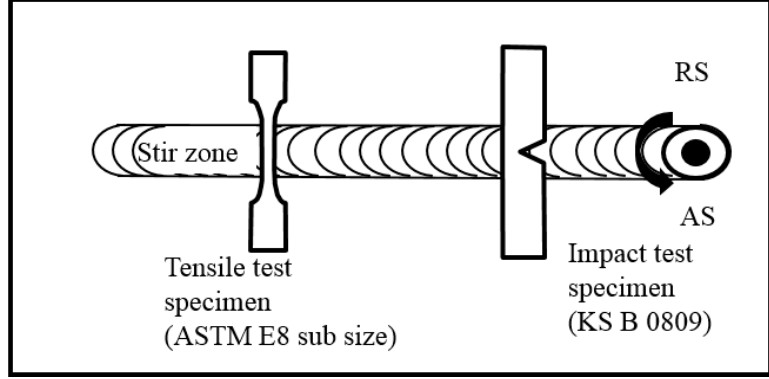

**Figure 2.** Schematic illustration of the friction-stir-welded plate and test specimen preparation.

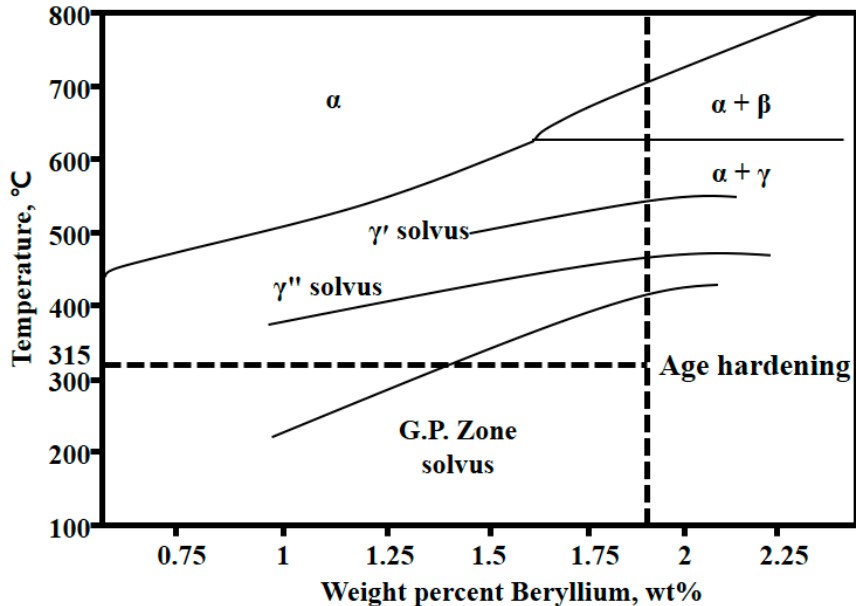

**Figure 3.** Phase diagram and metastable solvus of the beryllium-copper alloy.

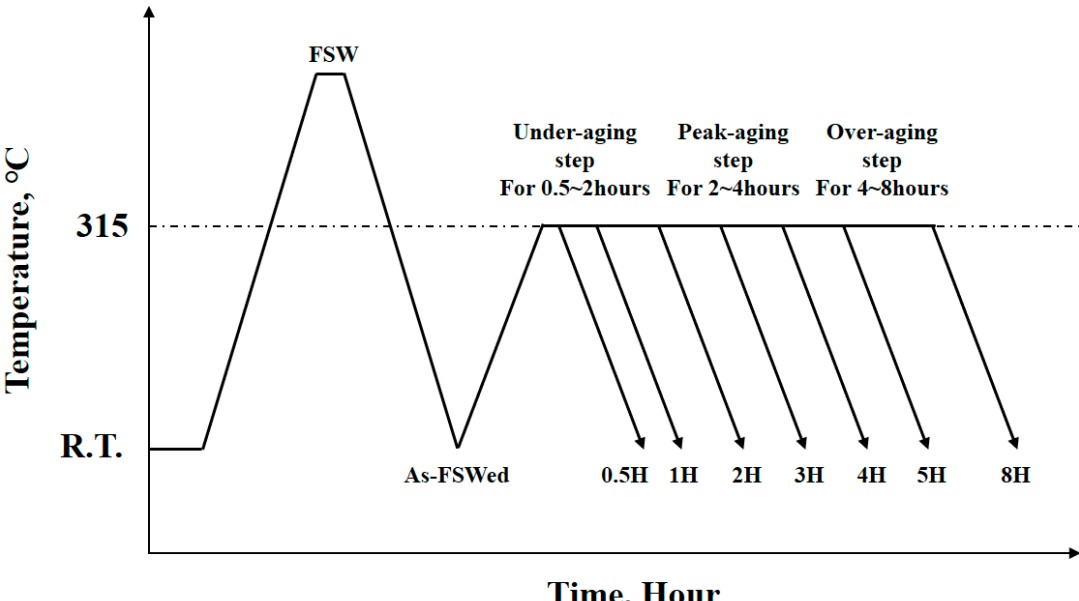

**Figure 4.** Process flow chart including the friction stir welding (FSW) and post-weld heat treatment conditions.

## 3. Results

### 3.1. Microstructure and Mechanical Properties of the FSW Joints and the Base Metal

The mechanical properties of the base metal and the joints are listed in Table 2. The tensile strength and hardness of the base metal were 1162 MPa and 380 HV, respectively, showing toughness values of 4.6 J and elongation of 9.4%. After FSW, both the tensile strength and hardness decreased down to 461 MPa and 162 HV, respectively. However, the toughness and elongation showed significant corresponding increases of 92.2 J and 53%. It was revealed that FSW has a considerable effect on the mechanical properties of beryllium-copper alloys implying that microstructural changes occur in the welding region during FSW. An optical micrograph and a TEM bright-field image of the base metal are shown in Figure 5a,b, respectively. From Figure 5a, it can be seen that the equiaxed α matrix ranged from 40 to 150 μm. In Figure 5b, a dense γ′ (CuBe) needle-like precipitates can be observed. Figure 6 shows the overall cross-section of the FSW joints. It can easily be identified from Figure 6a that the welding zone of the FSW joint is divided into three regions: the stir zone (SZ), the thermo-mechanical-affected zone (TMAZ) and the heat-affected zone (HAZ). The coarse grain structure in the base metal (BM) is replaced by introducing the SZ containing refined grains (as shown in Figure 6b). This is attributed to the dynamic recrystallization caused by severe thermal plastic deformation in the SZ. A mixture of coarsened, refined and elongated grains in the TMAZ (a transitional region between the SZ and the HAZ) was typical due to the incomplete dynamic recovery and recrystallization in this zone (Figure 6c). The grain size in the HAZ (Figure 6d) was similar than that of the BM (Figure 5a). Figure 7 shows the microstructures of the BM and SZ. Compared with the BM (Figure 7a) and the SZ (Figure 7c) under low magnification, it is apparent that the grains are significantly refined due to the dynamic recrystallization stemming from the frictional heat and plastic flow of the softened material during FSW. It could also be observed that the microstructure consisted of discontinuous precipitation (DP) cells (γ) located at the grain boundaries of α matrix, as presented in Figure 7b. As shown in Figure 7d, the grain boundaries in the SZ do not have DP cells due to their dissolution into the BM during FSW. Figure 8 shows a TEM bright-field image of the SZ. The microstructure consists of only α phase, without γ′ precipitates. This suggests that the strengthening γ′ precipitates dissolved into the BM during the FSW heat cycle. Such dissolution of the γ′ precipitates can be related to decreases in the hardness and tensile strength of the SZ.

**Table 2.** Mechanical properties of the base metal and the FSW joints.

| Specimen | Tensile Strength (MPa) | Yield Strength (MPa) | Failure Strain (%) | Hardness (HV) | Toughness (J) |
|---|---|---|---|---|---|
| Base metal | 1162 | 673 | 9.4 | 380 | 4.6 |
| FSW joints | 461 | 258 | 53 | 162 | 92.2 |

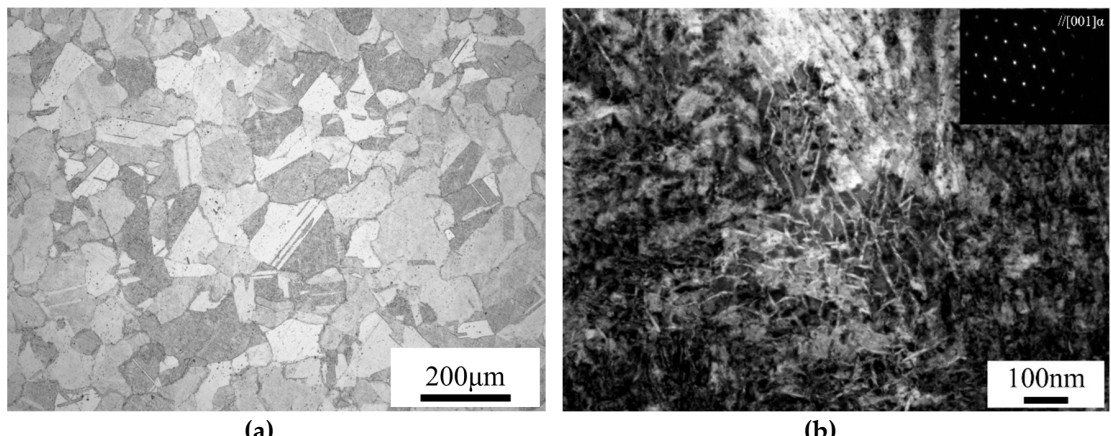

**Figure 5.** Microstructural analysis of the base metal: (**a**) optical microscopy, (**b**) transmission electron microscopy.

**Figure 6.** Cross-sectional macrograph of (**a**) the FSW region and (**b**) micrographs of the stir zone (SZ), (**c**) thermo-mechanical-affected zone (TMAZ) and (**d**) heat-affected zone (HAZ).

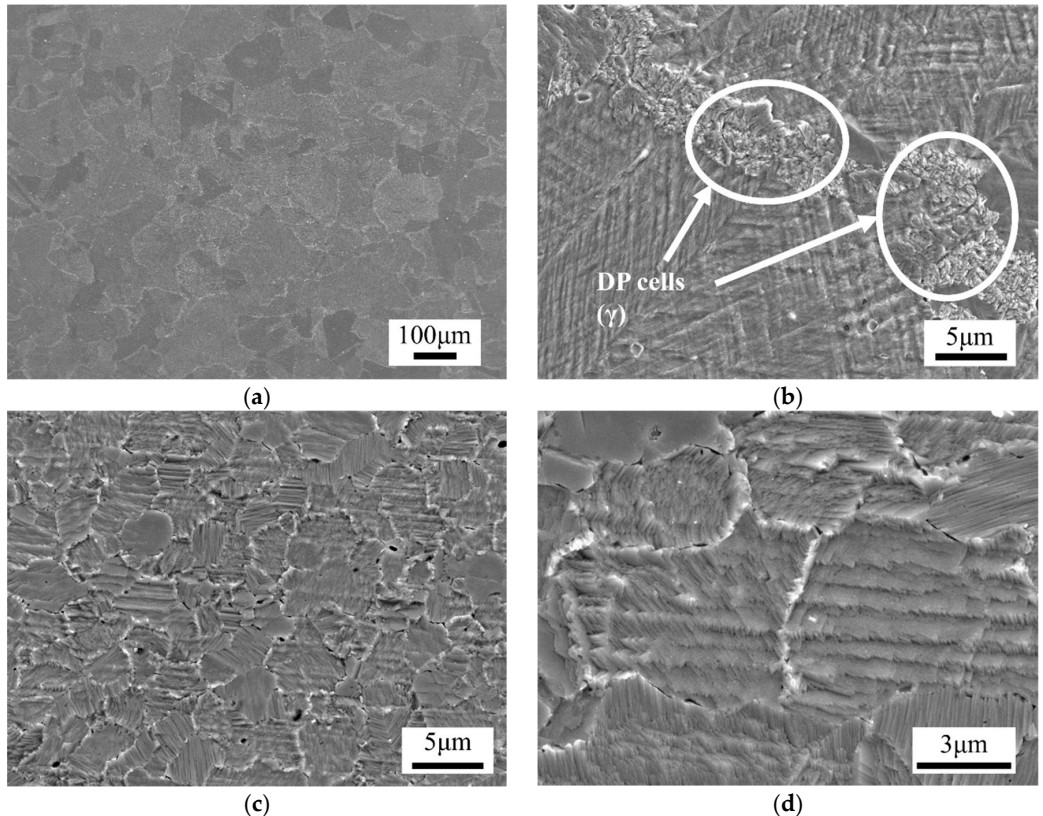

**Figure 7.** Scanning electron microscope (SEM) Images of the base metal: (**a**) 100×, (**b**) 3000× and stir zone (**c**) 3000×, and (**d**) 7000×.

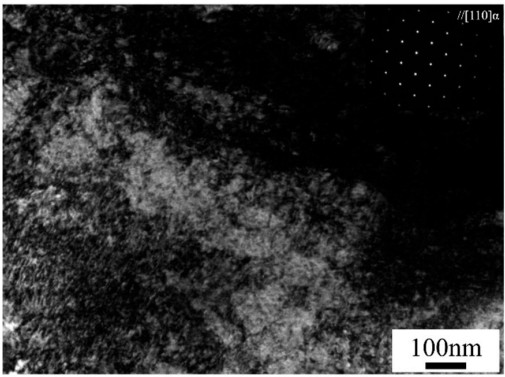

**Figure 8.** Transmission electron microscope (TEM) bright-field images in the stir zone.

*3.2. Behavior of Mechanical Properties during PWHT*

Figure 9 presents the outcomes of tests of the Vickers hardness and Charpy impact absorption energy in the stir zones of FSW joints subjected to a post-weld heat treatment. When the FSW was performed, the hardness decreased significantly to 162 HV from 380 HV. This provides evidence that the hardness was decreased in the SZ due to the dissolution of the strengthening precipitates, as shown in Figure 8. When the joint was subjected to PWHT at 315 °C, the microhardness increased steeply during the under-aging step (US) of the PWHT process. Furthermore, as PWHT lasted for approximately three hours, the results for the joint showed that the hardness values recovered similarly to the BM outcome. In other words, this was the peak-aging step (PS). When PWHT lasted for more than four hours, the hardness decreased slightly. Thus, this was the over-aging step (OS). The Charpy impact absorption energy increased up to 92.2 J from 4.6 J, as listed in Table 2, as the FSW was conducted. However, it decreased severely during the under-aging step. Subsequently, the decrease lasted for four

hours. However, the Charpy impact absorption energy value increased slightly during the over-aging step. Figure 10 shows the behavior of the tensile properties of the FSW joints after PWHT. All of the tensile test specimens were fractured at the BM. Both the tensile and yield strength increased early during the under-aging step. Subsequently, these increases lasted for four hours of PWHT. After the peak-aging step, these values decreased gradually. On the other hand, the strain decreased abruptly early in the under-aging step. Additionally, it showed little change during the peak-aging and over-aging steps. Hence, PWHT has a decisive effect on the recovery of mechanical properties such as the hardness, yield strength and tensile strength of FSW joints. These results also indicate that microstructural evolution such as precipitation can occur during PWHT.

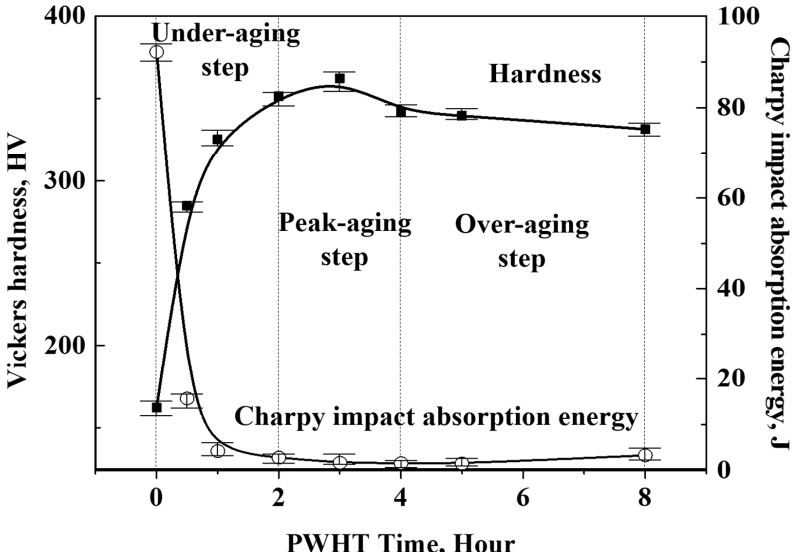

**Figure 9.** Behavior of the Vickers hardness and Charpy impact absorption energy in the stir zone of FSW joints subjected to a post-weld heat treatment. PWHT: post-weld heat treatment.

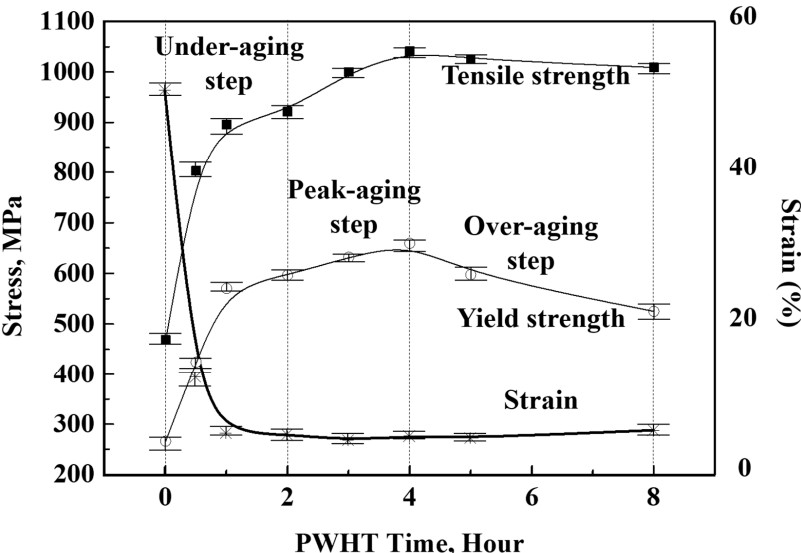

**Figure 10.** Behavior of the tensile properties of FSW joints after a post-weld heat treatment.

### 3.3. Microstructural Evolution during PWHT

Figure 11 shows SEM images of DP cells formed at the grain boundaries of the stir zone after the samples were subjected to PWHT for 30 min, three hours and eight hours. In the early stage of the under-aging step, very few and relatively small (Figure 11a) DP cells were observed. When the

PWHT process was extended to three hours, the number of DP cells increased (Figure 11b). After eight hours, many colonies of DP cells formed (Figure 11c). The DP cells are involved in the formation of a solute-depleted matrix phase ($\alpha'$) and a precipitate phase ($\gamma$) as a duplex transformation product, typically with nucleation at the grain boundaries with growth into one side of the supersaturated matrix ($\alpha$ phase) [22–25]. The DP cells consumed numerous solute atoms and diminished the precipitation hardenability; in this case, the $\gamma$ phase readily formed instead of the finer metastable $\gamma'$ and $\gamma''$ phases. TEM observations were used to characterize the precipitates which formed during PWHT, and these results are shown in Figure 12. As shown in Figure 8, no precipitates were found at the foil prepared from the FSW joints. However, numerous nano-scale sphere-type precipitates were observed in the foil prepared from the sample which underwent PWHT for 30 min (Figure 12a). Many $\gamma'$ phase precipitates arose depending on the duration of PWHT (Figure 12b). When the PWHT process lasted for eight hours, coarsened $\gamma'$ phases were observed (Figure 12c).

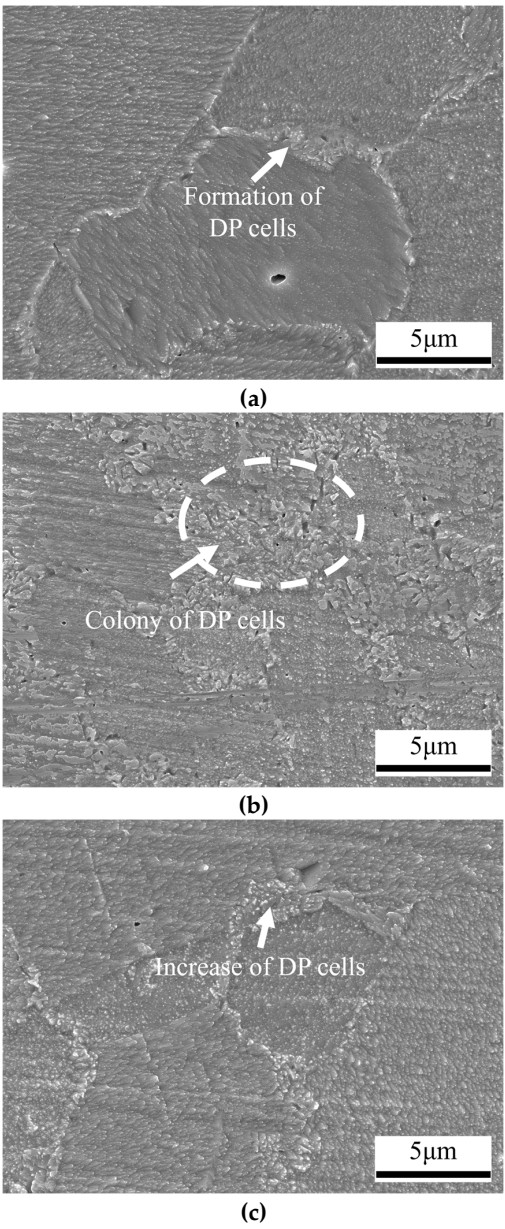

**Figure 11.** SEM images of discontinuous precipitation (DP) cells formed at the grain boundaries of the stir zone subject to PWHT: (**a**) 0.5 hour, (**b**) three hours and (**c**) eight hours.

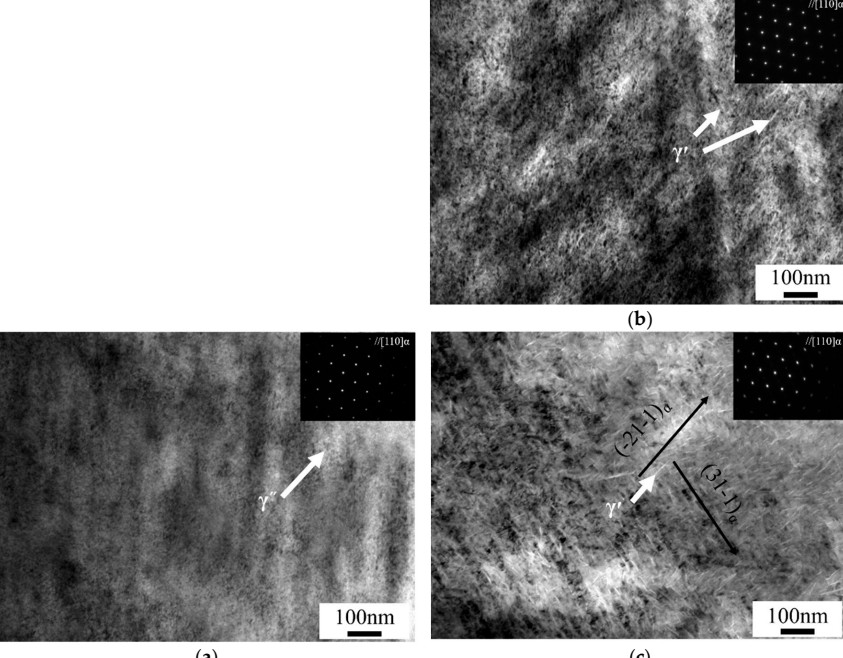

**Figure 12.** High-resolution transmission electron microscope (HR-TEM) images and selected area diffraction pattern (SADP) outcomes of the post-weld heat treated foils: (**a**) 0.5 hour, (**b**) three hours and (**c**) eight hours.

Figure 13 shows high-magnification CS-TEM bright-field images and corresponding SAD patterns of samples after PWHT for 30 min (a) and three hours (b) with the axis parallel to $[001]_\alpha$. In Figure 13a, the $\gamma''$ precipitates were nucleated and grew to a length of 5~15 nm along the $(-100)_\alpha$ direction with three to six layers of Be. With a PWHT duration up to three hours, the $\gamma''$ precipitates grew and coarsened to $\gamma'$ in the same direction and more than eight Be layers formed. The images indicate the distance between the atoms with a straight line in two directions, showing results of a = 0.236 nm, b = 0.232 nm (Figure 13a), and a = 0.270 nm, b = 0.279 nm (Figure 13b). These outcomes were fairly close to the phase parameter of $\gamma'$ as reported by Guoliang et al. [21].

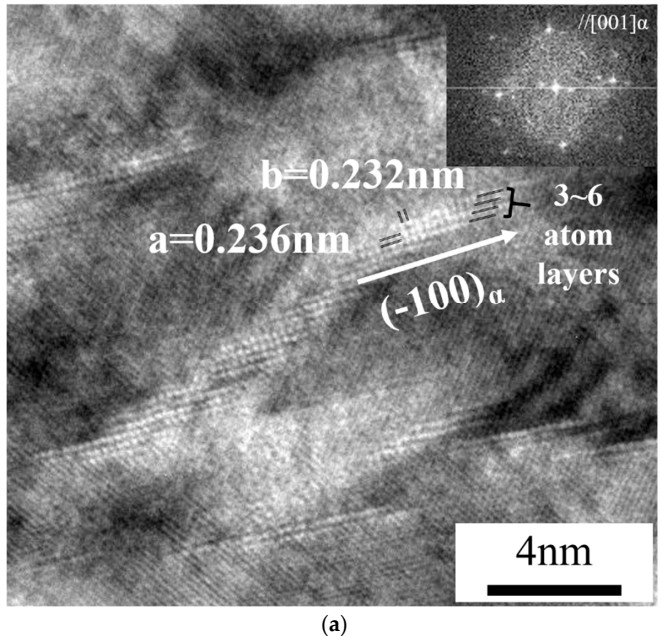

(**a**)

**Figure 13.** *Cont.*

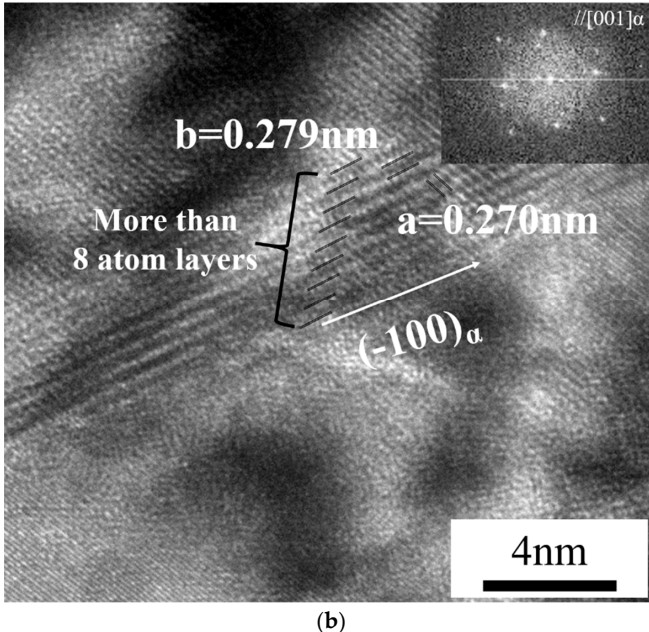

(**b**)

**Figure 13.** CS-TEM images and SADP outcomes of precipitates consisting of (**a**) $\gamma''$ and (**b**) $\gamma'$ phases in PWHTed foils.

## 4. Discussion

*Relationship between the Mechanical Properties and Microstructure during FSW and Consequent PWHT*

In case of laser beam welding butt joint of beryllium-copper alloy, the joint shows liquation crack in HAZ involved in fracture. Also it is possible to get sound joint properties of 0.2 mm of thickness plate after welding; however, there is currently limited research on more than 3 mm of thickness [9].

In the case of solid-state joining process, which use not only thermal factor but secondary factors such as mechanical or chemical factor, the problems caused by solidification from melting state is expected to be suppressed.

Esmati et al. attempted diffusion brazing lap joint of beryllium copper alloy with filler metal of Ag content. The brazing joint was successfully fabricated without defect at 750 °C for 1200 s. However, maximum tensile strength was comparatively low value as 173 MPa [10].

Especially, this study was conducted to get a sound thick plate joint that has good joint properties without solidification defects by proposing the properly controlled friction stir welding condition.

Based on preliminary welding test in several friction stir welding conditions, the authors were able to figure out optimal condition to control the heat input properly. When the welding condition accompanied higher heat input, since the microstructure of the joint is expected to exhibit greater dissolution of the $\gamma'$ and $\gamma$ precipitates, the faster and the stronger hardening effect is expected in the same post-weld heat treatment condition due to the higher driving force. Meanwhile, when the heat input during the welding process was lower, since the less $\gamma'$ and $\gamma$ precipitates are dissolved, it is assumed to show the slower and the weaker hardening effect due to the less driving force to form precipitates.

The effect of a post-weld heat treatment on the mechanical properties and microstructures of friction-stir-welded beryllium-copper alloy were investigated, as presented in the results section. In this section, the relationship between the mechanical properties and precipitation behavior during FSW and consequent PWHT is discussed. The process flow from BM to PWHT has five steps based on the alternation of the mechanical properties and microstructures. This process is shown in Figure 14. Initially, the base metal has acceptable hardness and tensile strength, as a considerable amount of $\gamma'$ precipitates, which are the primary strengthening mechanism of the present alloy, were present in the BM, as shown in Figure 5b. Interestingly, in the second step just after FSW, the tensile strength and

hardness decreased abruptly, while the toughness and ductility increased sharply. The non-existing presence of strengthening $\gamma'$ precipitates is considered to be the primary cause of the fall in both the hardness and tensile strength. The authors believe that the $\gamma'$ precipitates dissolved into the base metal owing to the frictional heat generated by the rotation of the tool. A significantly refined grain microstructure was formed in the stir zone. Both the refined grains and softening through the dissolution of the $\gamma'$ precipitates had an effect on the upsurge of the toughness. The PWHT period includes steps 3, 4 and 5. During the third step, called the under-aging stage of PWHT within a half hour (PWHT, US), a recovery of the hardness and tensile strength was obtained. In contrast, the toughness and ductility expressed a large decline. This outcome appears to be related mainly to the formation of globular $\gamma''$ precipitates, as shown in Figures 12a and 13a. During the fourth stage of the peak-aging step of PWHT from two hours to four hours (PWHT, PS), the gradual strengthening tendency continues as the aging time is extended. The peak tensile strength and hardness could be determined from microstructures mainly containing $\gamma'$ precipitates which grew from $\gamma''$ precipitates. Finally, during the fifth stage of the over-aging step of PWHT from four hours to eight hours, a slight decrease in hardness and tensile strength occurred. The growth and coarsening of $\gamma'$ precipitates are considered to be the dominant causes of this finding. If PWHT involves longer aging times up to a few days, the low mechanical properties through the softening of the microstructure are suggested to appear. It was clearly revealed that when the Giga-grade high-strength beryllium-copper alloy, which shows mechanical properties mainly due to precipitation of the $\gamma'$ phase is subjected to FSW, the hardness and tensile strength decrease sharply because the $\gamma'$ precipitates dissolve into the base metal due to the frictional heat generated during the FSW process. It was also found that PWHT is indispensable to recover the hardness and tensile strength of FSW joints. However, excess aging times exceeding three to four hours at 315 °C bring about a decline in the hardness and tensile strength. In conclusion, it is crucial to control metastable precipitates such as $\gamma''$ and $\gamma'$ to secure reliable FSW joints with beryllium-copper alloys.

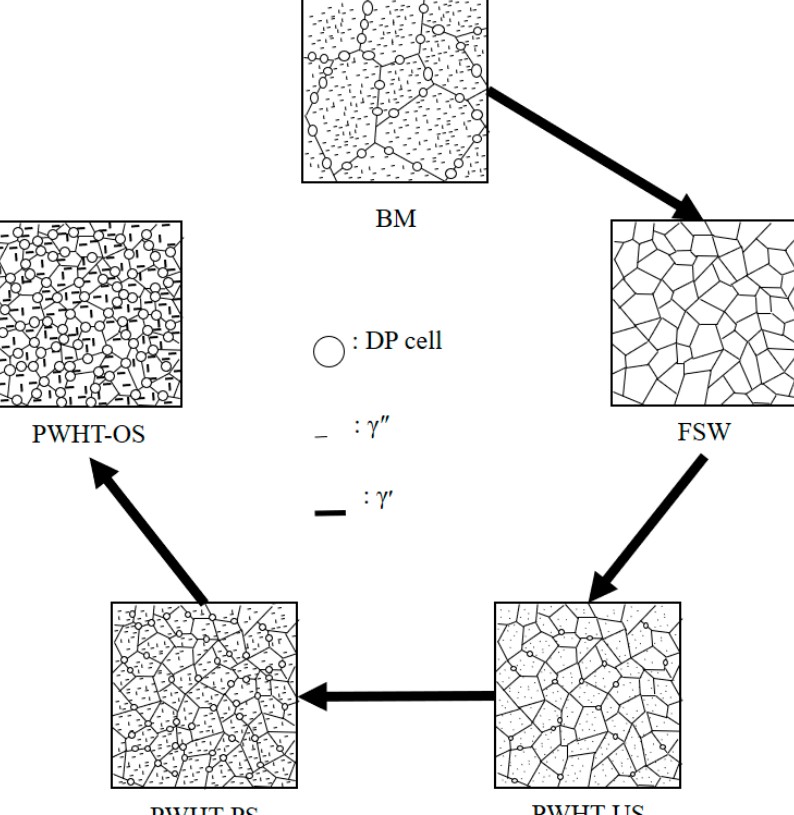

**Figure 14.** Schematic diagram of the evolution of the microstructure during the FSW and PWHT processes.

## 5. Conclusions

To understand the microstructural evolution and behavior of the mechanical properties during FSW with a consequent PWHT process, FSW was conducted using a beryllium-copper alloy and the resulting joints were subjected to PWHT at 315 °C for up to eight hours. The authors determined the microstructural and mechanical behaviors and the relationship between both outcomes. The results from this study are summarized below.

1. Friction stir welding of a Giga-grade high-strength beryllium-copper alloy was successfully conducted with a tool rotational speed and travel speed of 700 RPM and 60 mm/min, respectively. Sound joints without defects could be obtained.
2. A refined grain microstructure was formed in the stir zone. The strengthening $\gamma'$ (CuBe) precipitates dissolved into the base metal due to the frictional heat generated during the FSW process.
3. After FSW, the hardness and tensile strength decreased significantly, whereas the toughness and ductility increased sharply. It was found that the dissolution of $\gamma'$ precipitates is the dominant cause of the mechanical property changes during FSW.
4. When PWHT is conducted with the FSW joints, $\gamma''$ precipitates forms at an early stage within a half hour during the under-aging step. As the PWHT process is maintained, $\gamma'$ precipitates, which are the primary strengthening phase of this alloy, forms in the stir zone. This increased with an increase in the PWHT time. They became coarsened when the PWHT process exceeded four hours.
5. When the FSW joints are subjected to PWHT, the hardness and tensile strength increase remarkably at an early stage of the under-aging step. The gradual increase is maintained for up to four hours of PWHT. Past this time, however, the hardness and tensile strength gradually decrease. The toughness and ductility expressed crosscurrents during the same PWHT process.
6. To conclude, to obtain sound joints with acceptable mechanical properties, PWHT of 3~4 h at 315 °C is essential and the precipitation of the $\gamma'$ phases must be controlled.

**Author Contributions:** Conceptualization, K.L.; Investigation, Y.L.; Project administration, K.L.; Validation, S.M.; Writing–original draft, Y.L.; Writing–review & editing, K.L.

**Funding:** This work was financially supported by R&D program of the Korea Institute of Industrial Technology (JA180012). The authors would like to express their gratitude to KITECH.

**Acknowledgments:** The FSW tools used in this study were supported by R&D program of Ministry of Trade, Industry and Energy (NK180051). The authors would like to express their gratitude to MOTIE.

**Conflicts of Interest:** The authors declare no conflict of interest.

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
