# Peer review of "Effects of a Post-Weld Heat Treatment on the Mechanical Properties and Microstructure of a Friction-Stir-Welded Beryllium-Copper Alloy"

_metals, doi:10.3390/met9040461_

Round 1

Reviewer 1 Report

Overall, I found the manuscript a bit difficult to follow so I list some suggestions to make it easier to read, accept at your best convenience.

Line 13. What does WC stand for?

Line 16. As you performed treatments for up to eight hours instead this should be clearly described.

Line 17. were observed...

18. Vickers hardness, Charpy and tensile testing.

19. Observed softening … is suggested to be due to....

23. And the coarse gamma prime.... 

49-52. Unclear meaning, please read again and correct.

49, In recent days...

73. shows schematic representations of tensile...

74. Standard tensile test? or testing following ASTM:E8 standard?

Make sure you use either Celsius or Kelvin. Same situation for time units.

84. ...observed by OM using a Olympus....86. Surfaces were observed after...

89. Precipitation behaviour?

94. ...a cutting machine. Redundant!

99. replace cross head with strain rate.

100. 0.2 what? Please add units or percentage.

Sub size? Please add more details of tensile samples. Gauge length for example.

Figures 2 and 3 are in Celsius and discussion in kelvin, please correct.

117. respectively, showing toughness values of 4.6J...

119. ...decreased down to 461MPa...

121. ...alloys implying that...

125. ...needle -like precipitates can be observed... 

128. ...by introducing the SZ zone containing refined grains...

131. ...in the TMAZ (a transitional region...)...

133. ...was similar than that...

141. … without gamma prime precipitates. This suggests...

143. ...cycle. Such dissolution of the gamma prime precipitates can be related to decreases...

Table 2 and figure 9 Strain? what type? Failure strain?

181. energy decreased down to...

255. … were investigated...

259. … has acceptable hardness...

261. … were present in the BM

263-264 has unclear meaning, did the authors wanted to say that "the non-existing presence of"...

268. … third step called under-aging...

269. … a recovery...

279. … the low mechanical properties.... microstructure are suggested to appear.

280. ...alloy, which shows mechanical properties mainly due to precipitation of the gamma prime phase,...

Author Response

The authors would like to great thank to your kind and helpful advice. Please check the modified manuscript.

Line13. What does WC stand for?

è WC means tungsten carbide. The authors edited the sentence from ‘WC’ to ‘tungsten carbide (WC).

Line16. As you performed treatments for up to eight hours instead this should be clearly described.

è The authors modified the sentence from ‘eight hours’ to ‘a half, one, two, three, four, five and eight hours, respectively’.

Line17. were observed...

è The authors modified the sentence on a reviewer’s advice.

Line18. Vickers hardness, Charpy and tensile testing.

è The authors modified the sentence on a reviewer’s advice.

19. Observed softening … is suggested to be due to....

è The authors modified the sentence on a reviewer’s advice.

23. And the coarse gamma prime.... 

è The authors modified the sentence on a reviewer’s advice.

49-52. Unclear meaning, please read again and correct.

è The authors modified and completed the sentences on a reviewer’s advice.

49, In recent days...

è The authors modified the sentence on a reviewer’s advice.

73. shows schematic representations of tensile...

è The authors modified the sentence on a reviewer’s advice.

74. Standard tensile test? or testing following ASTM:E8 standard?

Make sure you use either Celsius or Kelvin. Same situation for time units.

è The authors modified the sentence on a reviewer’s advice.

84. ...observed by OM using a Olympus....

è The authors modified the sentence on a reviewer’s advice.

86. Surfaces were observed after...

è The authors modified the sentence on a reviewer’s advice.

89. Precipitation behaviour?

è The authors modified the sentence on a reviewer’s advice.

94. ...a cutting machine. Redundant!

è The authors deleted ‘a cutting machine’ in the sentence on a reviewer’s advice.

99. replace cross head with strain rate.

è The authors modified the sentence on a reviewer’s advice.

100. 0.2 what? Please add units or percentage.

Sub size? Please add more details of tensile samples. Gauge length for example.

Figures 2 and 3 are in Celsius and discussion in kelvin, please correct.

è The authors inserted ‘percentage’ and gauge length of ASTM E8 sub size specimen to the sentence on a reviewer’s advice. Unified temperature unit by Celsius

117. respectively, showing toughness values of 4.6J...

è The authors modified the sentence on a reviewer’s advice.

119. ...decreased down to 461MPa...

è The authors modified the sentence on a reviewer’s advice.

121. ...alloys implying that...

è The authors modified the sentence on a reviewer’s advice.

125. ...needle -like precipitates can be observed... 

è The authors modified the sentence on a reviewer’s advice.

128. ...by introducing the SZ zone containing refined grains...

è The authors modified the sentence on a reviewer’s advice.

131. ...in the TMAZ (a transitional region...)...

è The authors modified the sentence on a reviewer’s advice.

133. ...was similar than that...

è The authors modified the sentence on a reviewer’s advice.

141. … without gamma prime precipitates. This suggests...

è The authors modified the sentence on a reviewer’s advice.

143. ...cycle. Such dissolution of the gamma prime precipitates can be related to decreases...

è The authors modified the sentence on a reviewer’s advice.

Table 2 and figure 9 Strain? what type? Failure strain?

è Yes, it means failure strain. The authors modified on a reviewer’s advice.

181. energy decreased down to...

è The authors think this is reviewer’s misunderstanding. The authors modified the sentence for easy understanding. (Line 188~189)

è The authors found mistake in Fig. 8. They corrected the range of Charpy impact absorption energy. (Fig. 8)

255. … were investigated...

è The authors modified the sentence on a reviewer’s advice.

259. … has acceptable hardness...

è The authors modified the sentence on a reviewer’s advice.

261. … were present in the BM

è The authors modified the sentence on a reviewer’s advice.

263-264 has unclear meaning, did the authors wanted to say that "the non-existing presence of"...

è Yes, the authors modified the sentence on a reviewer’s advice.

268. … third step called under-aging...

è The authors modified the sentence on a reviewer’s advice.

269. … a recovery...

è The authors modified the sentence on a reviewer’s advice.

279. … the low mechanical properties.... microstructure are suggested to appear.

è The authors modified the sentence on a reviewer’s advice.

280. ...alloy, which shows mechanical properties mainly due to precipitation of the gamma prime phase,...

è The authors modified the sentence on a reviewer’s advice.

Reviewer 2 Report

This paper investigated the mechanical properties and microstructure of an FSW joint of a beryllium-copper alloy. Samples of as-welded and PWHT was analyzed. It was concluded that good quality of joints was obtained with a tool rotational speed of 700RPM and tool travel speed 14 of 60mm/min. The paper reads well and definitely advances the current understanding of the FSW process. However, before publication, the following needs to be considered:

In the abstract, please provide the full form of WC.

In the abstract delete the sentence, "The mechanical properties were measured by a Vickers 18 hardness test, a Charpy impact test and a tensile test." It is not appropriate in the abstract.

Please avoid grouped references [8-13], [14-17], [20-22]. You should give proper credit to the previous investigator by writing in one/two sentences what they found in each reference.

 The last paragraph of the introduction should be rewritten emphasizing on why the reader should read your paper; what was the research gap and what needs to be studied.

Page 3: The results are from an average of 5 samples. I  would suggest adding the errors in Table 2.

Page 5 line 137: please check with the figures for α vs γ. In the figure, it shows γ.

Please include the legends in figures 8 and 9.

page 8 line 181: Check the values for energy absorption. the text description does not match with the figure.

Author Response

The authors would like to thank for your useful advice. Please confirm the modified manuscript.

In the abstract, please provide the full form of WC.

è The authors modified the sentence from ‘WC’ to ‘tungsten carbide (WC)’ on a reviewer’s advice.

In the abstract delete the sentence, "The mechanical properties were measured by a Vickers 18 hardness test, a Charpy impact test and a tensile test." It is not appropriate in the abstract.

è The authors delete the sentence on a reviewer’s advice.

Please avoid grouped references [8-13], [14-17], [20-22]. You should give proper credit to the previous investigator by writing in one/two sentences what they found in each reference.

è The authors avoided grouped references and mentioned briefly what the previous investigators found in each reference on a reviewer’s comment.

 The last paragraph of the introduction should be rewritten emphasizing on why the reader should read your paper; what was the research gap and what needs to be studied.

è The authors modified the paragraph on a reviewer’s comment.

Page 3: The results are from an average of 5 samples. I would suggest adding the errors in Table 2.

è The authors would like to ask for kind understanding about maintaining Table 2. All of values including minimum and maximum are very similar to the average value. Those are not the errors.

Page 5 line 137: please check with the figures for α vs γ. In the figure, it shows γ.

è The authors modified the sentence on a reviewer’s comment.

Please include the legends in figures 8 and 9.

è The authors would like to ask for kind understanding about maintaining figure 8 and 9. Because both figures already have explain for the legends. It is considered that readers could easily understand.

page 8 line 181: Check the values for energy absorption. the text description does not match with the figure.

è The authors modified the sentence for easy understanding.

è The authors found mistake in Fig. 8. They corrected the range of Charpy impact absorption energy.

Reviewer 3 Report

The paper investigates the influence of different PWHT routines on the mechanical properties of FSW welded beryllium-copper alloy.

The paper can be published after thorough revision of the following aspects:

- The introduction is very general. This part should answer the following questions: What was the main motivation? How is the state of the art? What is the progress in the field investigated by others? Plus, avoid multi-referencing (e.g. [14-17, 20-22]), instead please refer to each single reference and its specific contribution.

- If this is important to your study, a picture of the FSW tool could be added.

- The investigation routine is suited for the evaluation of the influence of the PWHT routine for one single set of welding parameters, but the main outcome is missing in the paper. What is the result? Which PWHT is recommended for this specific alloy and the used welding conditions. What happens to the mechanical properties when the welding parameters are changed?

- Avoid numbering of single sub-sections (4.1).

- The picture quality is poor in many cases, if this is due to compressing the pics, please upload high resolution figures.

- Add a blank space between numbers and units.

Author Response

The authors would like to great thank to your useful advice. Please check the modified manuscript.

- The introduction is very general. This part should answer the following questions: What was the main motivation? How is the state of the art? What is the progress in the field investigated by others? Plus, avoid multi-referencing (e.g. [14-17, 20-22]), instead please refer to each single reference and its specific contribution.

è The authors modified introduction part. We emphasized on why the reader should read this paper and what’s the purpose and new knowledge, on a reviewer’s advice.

è The authors avoided grouped references and mentioned briefly what the previous investigators found in each reference on a reviewer’s comment.

- If this is important to your study, a picture of the FSW tool could be added.

è The authors do not want to add a picture of the FSW tool.

- The investigation routine is suited for the evaluation of the influence of the PWHT routine for one single set of welding parameters, but the main outcome is missing in the paper. What is the result? Which PWHT is recommended for this specific alloy and the used welding conditions. What happens to the mechanical properties when the welding parameters are changed?

è The authors described the optimum PWHT condition to the conclusion (number 6) on a reviewer’s advice.

è The authors think the 2nd question is an expanded issue, not an intrinsic issue of the present paper. As a matter of fact, the authors now do that to get the answers and want to submit a separate paper if you’ll let us. The authors would like to ask for kind understanding for this situation.

- Avoid numbering of single sub-sections (4.1).

è The authors deleted the number ‘4.1’.

- The picture quality is poor in many cases, if this is due to compressing the pics, please upload high resolution figures.

è The authors tried to improve the figure quality and uploaded high resolution figures.

- Add a blank space between numbers and units.

è The authors add a blank space between numbers and units.

Round 2

Reviewer 3 Report

The manuscript did not improve significantly compared to the first submission.

- The introduction part remains very general. The multi-references were replaced by single references, but the motivation for the investigation and the research gap were not addressed properly. The state of the art for the selected special alloy was also not reflected.

- The tool geometry remains unclear, but this is essential for the heat generation in FSW. In other words, the investigation is not reproducible by others.

- As other reviewer stated correctly, error bars are missing where needed.

- Not one single reference was used in the discussion part, so the discussion and comparison of the results of this investigation with the state of the art is missing.

- From my point of view, only one single set of welding parameters is not sufficient to give recommendations for the PWHT routine.

Author Response

The authors would like to great thank to your useful advice. Please check the modified manuscript.

- The introduction part remains very general. The multi-references were replaced by single references, but the motivation for the investigation and the research gap were not addressed properly. The state of the art for the selected special alloy was also not reflected.

è The authors modified introduction part. We included the investigation, the research gap and the state of the art, on a reviewer’s advice (line 42-50).

- The tool geometry remains unclear, but this is essential for the heat generation in FSW. In other words, the investigation is not reproducible by others.

è The authors edited the illustration of tool geometry in materials and methods part.

- As other reviewer stated correctly, error bars are missing where needed.

è The authors included error bars in figure 9, figure 10.

- Not one single reference was used in the discussion part, so the discussion and comparison of the results of this investigation with the state of the art is missing.

è The authors modified discussion part (line 266-278) on a reviewer’s comment. We compared our research with beryllium-copper joints fabricated by laser beam welding and diffusion brazing method.

- From my point of view, only one single set of welding parameters is not sufficient to give recommendations for the PWHT routine.

è The authors modified discussion part (line 279-285). We emphasized on why was the properly controlled optimal condition adopted in this research.

Round 3

Reviewer 3 Report

Some of the comments of the last review were addressed in the revised version.

It remains still strange, that the newly added two (!) references in the discussion part, which should deserve as a standard level for comparison of the obtained results with the state of the art, were not cited in the introduction part. In other words, the introduction part is still incomplete.

The limitations if this preliminary study were addressed in the discussion part, but the very specific findings for a specific welding parameter set in this paper do not satisfy the very general title of the paper.

Author Response

The authors would like to great thank to your useful advice. Please check the modified manuscript.

- It remains still strange, that the newly added two (!) references in the discussion part, which should deserve as a standard level for comparison of the obtained results with the state of the art, were not cited in the introduction part. In other words, the introduction part is still incomplete.

è The authors modified the introduction part including references, on a reviewer’s advice

- The limitations if this preliminary study were addressed in the discussion part, but the very specific findings for a specific welding parameter set in this paper do not satisfy the very general title of the paper.

è The authors modified materials and methods part. We included the reason that the authors chose the specific welding condition precisely.